# Semantic Metadata Annotation Services in the Biomedical Domain—A Literature Review

**Julia Sasse** , **Johannes Darms and Juliane Fluck** *

ZB MED—Information Centre for Life Sciences, 50931 Cologne, Germany; sasse@zbmed.de (J.S.); darms@zbmed.de (J.D.)
* Correspondence: fluck@zbmed.de; Tel.: +49-228-73-603-51

**Abstract:** For all research data collected, data descriptions and information about the corresponding variables are essential for data analysis and reuse. To enable cross-study comparisons and analyses, semantic interoperability of metadata is one of the most important requirements. In the area of clinical and epidemiological studies, data collection instruments such as case report forms (CRFs), data dictionaries and questionnaires are critical for metadata collection. Even though data collection instruments are often created in a digital form, they are mostly not machine readable; i.e., they are not semantically coded. As a result, the comparison between data collection instruments is complex. The German project NFDI4Health is dedicated to the development of national research data infrastructure for personal health data, and as such searches for ways to enhance semantic interoperability. Retrospective integration of semantic codes into study metadata is important, as ongoing or completed studies contain valuable information. However, this is labor intensive and should be eased by software. To understand the market and find out what techniques and technologies support retrospective semantic annotation/enrichment of metadata, we conducted a literature review. In NFDI4Health, we identified basic requirements for semantic metadata annotation software in the biomedical field and in the context of the FAIR principles. Ten relevant software systems were summarized and aligned with those requirements. We concluded that despite active research on semantic annotation systems, no system meets all requirements. Consequently, further research and software development in this area is needed, as interoperability of data dictionaries, questionnaires and data collection tools is key to reusing and combining results from independent research studies.

**Keywords:** interoperability; FAIR data; semantic metadata; metadata enrichment; annotation service





## 1. Introduction

A central focus of research data management is the representation of data. This concerns the readability of the data, especially the semantic information. This should be unambiguous to humans but also to computers. For conducting clinical and epidemiological studies, for example, case report forms, data dictionaries and questionnaires play an important role. They are used either to collect information or to document existing data. They are created by people so that they can be easily consumed by peers. Documentation of available variables is read by scientists and needs to be understood before using the data in any analysis. Similarly, questionnaires are read and completed by people. Therefore, the questions are naturally chosen so that they are easy to understand by humans and can be answered clearly. Even though questionnaires are often in digital form, they are most often not machine readable or understandable. Frequently, the meanings of the question and answer choices are not coded in such a way that a computer can interpret them; i.e., they are not semantically coded. As a result, the data analysis is more complex than necessary and data integration is a major issue.

That is especially true for the integration of multiple datasets. To decide on the equality of two data items, the definitions have to be compared. Not surprisingly, free-text

descriptions can differ even in simple cases, so that often only humans can decide whether they match. As a simple example, consider the many ways to ask about and describe a person's gender. e.g., "Are you male or female?"; "What gender are you?"; "What is your gender?". To overcome this challenge, semantic codes can be used to encode the meanings of a respondent variable and response options. When creating new questionnaires, one can code this information directly [1]. In the present example, gender could be coded with the SNOMED concept "Finding related to biological sex" (SCTID: 429019009) and the response options with SCTID: 248153007 (male) and SCTID: 248152002 (female). Such semantic enrichment makes case report forms, data dictionaries and questionnaires more interoperable and consequently reusable. This is an important step toward making data FAIR (Findable, Accessible, Interoperable, Reusable) [2] and ultimately allows them to be used outside of their original purpose. For example, semantic enrichment makes it possible to match different data collections. Such an alignment means that the data collected can be compared and combined, enabling cross-study analysis.

The integration of semantic codes into newly designed questionnaires is already supported by some tools (e.g., secutrial [3]; redcap [4]). Retrospective harmonization of metadata is a current topic in various research projects. The EU-Innovative Medicine project EHDEN [5] is organizing open calls for data partners to obtain financial support to standardize their data according to the OMOP Common Data Model. Under EHDEN, small and medium-sized enterprises (SMEs) are trained and funded for supporting data transformation.

The Maelstrom project [6], for example, uses approaches to semantically annotate data to make research data more FAIR. A Maelstrom target is to foster population-based cohort data discovery [7]. Guidelines to harmonize epidemiological research data have been published [8] and a study metadata model and the Maelstrom classification composed of 18 domains and 135 subdomains have been created. Data annotations are performed by experts and validated with an in-house automated classifier based on a machine learning method [7]. As a result, harmonized data can be searched via open-source software applications, such as OPAL and MICA [9].

Other examples include the German National Research Data Infrastructure (NFDI) [10] projects, which currently consist of 19 domain-specific projects that aim to optimize access to research data. One of these projects, the NFDI for Personalized Health Data [11], aims to improve the FAIR access to structured health data originating from epidemiology studies, public health and clinical studies [12]. One major task of all these NFDI projects is the harmonization of (meta)data. The assignment of such annotations to harmonized standards needs to be done by experts, but this is labor intensive and should ideally be facilitated by software systems. Therefore, the first step within NFDI4Health was to collect requirements and review which software can support this work. Even though we focus on NFDI4Health, we also consider the general applicability of this service in other NFDIs. As a result, this review:

- Formulates requirements for tools supporting retrospective semantic annotation/ enrichment of (meta)data, questionnaires and data dictionaries derived from epidemiological studies, public health and clinical trials (Section 2);
- Reports a systematic literature search to find available techniques, technologies and tools to support users in retrospective semantic annotation (Section 3);
- Gives an overview of the existing metadata annotation services we could identify (Section 4.1);
- Evaluates the existing metadata annotation services against the set of requirements we formulated (Section 4.2).

## 2. Definition and Requirements of a Semantic Metadata Annotation Service

Data harmonization refers to the process of annotating/enriching data elements with vocabulary semantic codes. Ideally, a community agreed standard or quasi standard is used. As a result, harmonized data are more interoperable, correspond to the "I" in FAIR

and allow the computer to "understand" the meaning of data. In this context, we define a metadata annotation service as a supporting tool to enrich data elements within a digital object (e.g., a data dictionary or questionnaire) with semantic codes. Figure 1 illustrates the process of semantic metadata annotation of data collection instruments.

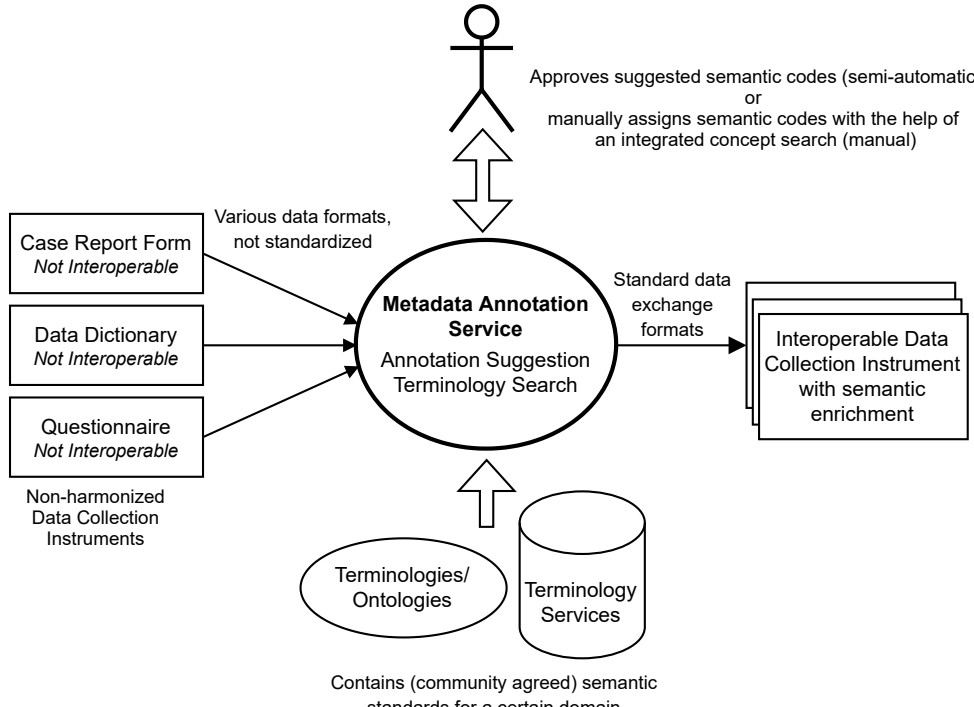

**Figure 1.** Data collection instruments are released in different standard and nonstandard formats. Without semantic annotation, they are not semantically interoperable. A metadata annotation service could facilitate semantic annotation and conversion to a standardized format. Semi-automatic support through annotation suggestions and integrated concept searching by providing interfaces to terminologies/ontologies and terminology services could help researchers overcome the challenges of this process.

Based on harmonization efforts published in the Maelstroem project and on use cases and a number of data annotation pilots addressed in NFDI4Health, we have gathered requirements for suitable software supporting this process. A typical harmonization use case in both projects would be the enrichment with semantic codes of all questions and response items in a questionnaire used in a public health survey.

In the following, we describe the challenges for a semantic metadata annotation service and the basic requirements that derive from the health and epidemiology domain, especially from the NFDI4Health project. Beyond NFDI4Health, we also consider open science and generalization aspects as important requirements. All identified requirements are summarized in Table 1 with short descriptions and a prioritization order we agreed upon in NFDI4Health.

The first requirement addresses openness and generalization of the tools. Since different formats are expected to some extent and adaptation of the software to a specific use case is required, the software must be available under an open-source license or should at least be openly accessible (Requirement 1).

A key feature is the combinability with existing, optimally specified and standardized data exchange formats. This is a mandatory requirement for a service that enables the annotation of data elements within a data object. Examples include the standard formats HL7 FHIR [13] and CDISC ODM [14] in the medical field. Therefore, different input and output formats must be supported or the technical interface must allow extension to other formats (Requirement 2).

The enrichment of metadata should follow FAIR principles. Consequently, the vocabularies and thus terminologies used for annotation must also be FAIR (Requirement 3), as stated in the FAIR Interoperability Principle: "(Meta)data use vocabularies that follow the FAIR principles" (Wilkinson et al. 2016) [2].

Two other requirements (Requirements 4 and 5) relate to the functionality of the system. Annotation of data elements is labor intensive and often involves a large number of elements, so this process should be as simple as possible. As a result, simply releasing or rejecting autonomously created annotations (Requirement 5) is preferred to manually searching for the appropriate concept (Requirement 4). However, we consider the feature of searching for a concept more important than an autonomous recommendation system, and hence prioritized the former as a must and the latter as a recommendation.

In addition, the semantic classes that can be annotated should be configurable, as each specific use case may require its own set. This requirement can either mean that custom ontologies/terminologies can be used or integrated into the software, or that a connection to a terminology/ontology or SPARQL [15] service can be configured. We have listed both requirements (Requirements 6 and 7) independently, since the latter allows reuse of existing infrastructure.

**Table 1.** Requirements and their prioritization for a semantic metadata annotation service based on current use cases in the NFDI4Health project.

| No. | Requirement | Description | Prioritization |
|---|---|---|---|
| 1 | Open accessible and/or Open Code | The software is available under an open-source license or the service is free to use | **MUST** |
| 2 | Support of common data formats | The software supports common and domain specific import and export data formats | **MUST** |
| 3 | FAIR Terminologies | Integrated terminologies comply with the FAIR principles | **MUST** |
| 4 | Terminology Search | The software provides a possibility to search for concepts/classes within integrated terminologies/ontologies | **MUST** |
| 5 | Annotation Suggestion | The software provides suggestions for semantic concept annotations for each data element (semi-autonomous process) | **SHOULD** |
| 6 | Interface to external terminology/ontology services | The software offers the possibility to connect to an external terminology/ontology/SPARQL service | **SHOULD** |
| 7 | Extension of Terminologies/Ontologies | The software allows one to extend the default set of integrated terminologies/ontologies | **MAY** |

## 3. Methods—Systematic Literature Review

A literature search based on the PRISMA guidelines [16] was undertaken in PubMed and Google Scholar using the search phrases "semantic metadata annotation" and "semantic metadata enrichment" in PubMed and "biomedical semantic metadata annotation" and "biomedical semantic metadata enrichment" in Google Scholar. We added the term "biomedical" to explicitly add the context to the Google Scholar queries, which is implicit when using PubMed. Altogether, 244 articles were screened—all of the PubMed results (144) and the first 50 results of each Google Scholar search. The searches were performed without additional search criteria (for example, time or type of articles) for a broad literature

analysis. We limited the result set of Google Scholar due to a large amount of search results (in total 33.770 results at the time of search) without relevancy. Even the first 50 results added no significant new finding. The number of duplicates increased and the number of titles and abstracts to include decreased drastically. Search results between 50 and 80 accounted for no new items; thus, the search was limited to 50 records of Google Scholar.

In the next step, all duplicate articles were removed. To further remove irrelevant articles the titles were analyzed. Titles that contained the key words "semantic," "metadata" or "annotation" in combination or alone were included. However, the topic indicated by the title had to be in the context of metadata or text annotation. As a result, the majority (170 of 211) of articles were excluded. Further, articles (22) were excluded in the abstract screening. The exclusion criterion for this step was metadata annotation not being the main topic of the article. For example, the report "The Semantic Data Dictionary—An Approach for Describing and Annotating Data" by Rashid et al. 2020 [17], fulfills the inclusion criteria in the title screening with the key words "semantic data" and "annotating data," but the focus lies in the specification of a data dictionary for a formalized assignment of a semantic representation of data [17]. Screening of titles and abstracts was done non-restrictively with regard to the kind of data that were annotated. Articles about biomedical text annotation, such as "SIFR annotator: ontology-based semantic annotation of French biomedical text and clinical notes" from Tchechmedjiev et al. [18], were included in the content screening, along with articles such as "Ontology-based annotations and semantic relations in large-scale (epi)genomics data" from Galeota and Pelizzola [19], which do not deal with the annotation of metadata but with the annotation of other biomedical data.

The final step was content screening and assessment for eligibility. The final inclusion criterion for the review was the description and provision of a service for semantic metadata annotation. The service has to provide a function for associating metadata items with semantic concepts either from ontologies or from terminologies. The review was supplemented by a screening of referenced articles in the given papers. We found two ([20,21]) additional relevant articles trough this transitive closure. Another three reports ([22–24]) were included that are not yet published but were known through exchange with other working groups or via non-systematic literature search. The initial search phrases were varied with the additional keywords "harmonization," "standardization" and "semantic coding" in the non-systematic search. Since only one other service was found in the process, a systematic search using these keywords was not performed. The overall review process is depicted in Figure 2. It shows the individual steps performed with the number of removed articles in each step. A table with the full list of screened articles, and their exclusion or inclusion reasons are given as supplementary information.

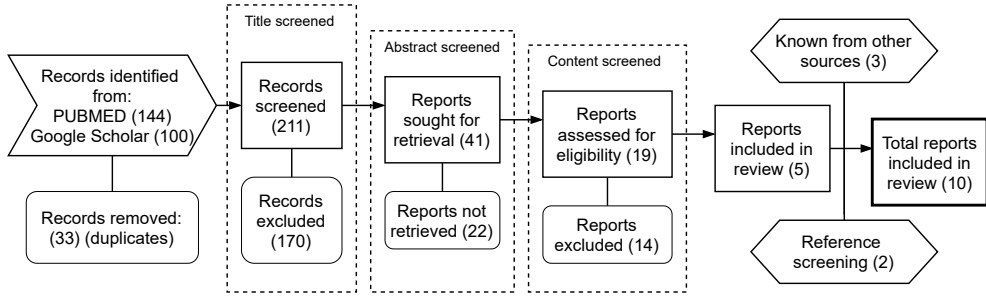

Search phrases: "semantic metadata annotation" (PUBMED), "semantic metadata enrichment" (PUBMED), "biomedical semantic metadata annotation" (Google Scholar), "biomedical semantic metadata enrichment" (Google Scholar)

**Figure 2.** The flow diagram illustrates the selection process at the various stages of the systematic review. It maps out the number of records identified from databases and included or excluded from the review. It is based on the PRISMA 2021 flow diagram [16].

*Categorization of Reviewed Articles*

During the review, we excluded many articles that are not about a software system for semantic enrichment in the biomedical field. We found that the articles we screened belong

to one of five classes. (1) ontology-based semantic annotation services, (2) metadata annotation services, (3) ontology design, (4) ontology services, and (5) named entity recognition methods for text mining.

Ontology-based semantic annotation services (1) are the main focus and subject of this survey and an overview of those services is given in Section 4.1. The other classes are described briefly, and examples of those services are given. Metadata annotation services (2) are services that help associate data objects with metadata, e.g., linking bibliographic information to a text document [25]. While these help to FAIRify data, they operate at the data object level. This is in contrast to group (1), which semantically enrich properties within an object (e.g., data dictionary or questionnaires).

Ontology design (3) describes a group of articles dealing with the creation of ontologies in specific fields or for specific purposes, as in [26] or [27].

Ontology services (4) such as the NCBO bioportal [28] or Ontology Lookup Service (OLS) [29] provide access to ontologies and terminologies and (5) named entity recognition methods for text mining provide techniques to find semantic concepts in unstructured or semi-structured documents [30,31]. In our understanding, both groups are parts or supporting services for a service of class (1), since they need to know which semantic concepts can be used for enrichment/annotation on the one hand, and on the other hand (2) they need to pair a text snippet within a data object with the corresponding semantic code.

## 4. Results

Here we present the results of our review. Each of the ten metadata annotation services we found is described in detail. Finally, we evaluate them against the defined basic requirements.

### 4.1. Existing Semantic Metadata Annotation Services

#### ODMedit

The Medical Data Models (MDM)-portal is an information infrastructure for medical research and health care and was developed in cooperation with the University and State Library of Münster. In addition to serving as a metadata registry, the system is used to create, analyze, share and reuse medical forms [32]. As part of the MDM-portal, ODMedit is a web-based application for creating data models with uniform semantic annotations for data integration in medicine. ODMedit mainly aims at the uniformity of annotations. Two matching data elements should have the same annotations. The basis for this is a public metadata repository. To achieve uniformity in encoding, semantic annotations can be reused if a matching data element exists in the metadata repository. In the semi-automated approach, annotations are suggested from the repository, which can then be selected by experts [33]. The forms are stored in the CDISC Operational Data Model (ODM) format [14]. The annotations are generated using the terminologies UMLS (Unified Modeling Language System) and SNOMED CT [32]. If no appropriate code is available, the user can follow links to the UMLS metathesaurus or the NCI metathesaurus websites. The web application can be used free of charge after registration.

#### Rightfield

RightField is a desktop application that allows embedding annotations using ontologies in Microsoft Excel [34] spreadsheet templates with data from life sciences [35]. Cells can be restricted to specific ranges of classes or instances from standard vocabularies. This allows users of spreadsheet templates to enter their data consistently without knowledge of the ontologies used. Annotations in RightField can be performed using ontologies from the NCBO BioPortal or using a local file in Web Ontology Language (OWL), Open Biological and Biomedical Ontologies (OBO), Resource Description Framework Schema (RDFS) and Resource Description Framework (RDF) formats [35]. The created template can be saved in CSV or RDF format. Rightfield is an open-access desktop application.

*Swate*

Swate is a Microsoft Excel add-in for semantic annotation of experimental data with controlled vocabularies developed in the context of the NFDI4Plants research project [23]. Basic features of Swate are gradual workflow annotation table building via basic annotation building blocks; terminology lookup with autocompletion, a relational term search to constrain the annotation column to a subset of ontologies; and advanced term search with multiple query parameters. No interface for integrating terminologies or terminology services is described, but the administrators of Swate can adapt the internal ontologies and can add concepts. Swate is open-access and currently integrates a customized set of more generic biological/chemical ontologies and plant-specific ontologies, for example Environment Ontology and Plant Ontology. As it is a Microsoft Excel add-in, import and export formats are Microsoft Excel files. Furthermore, CSV and RDF formats for export are available.

*OntoMaton*

OntoMaton provides ontology lookup and tagging services based on the NCBO Bioportal and the Annotator Web Service [21], the EMBL-EBI Ontology Lookup Service [29] and the Linked Open Vocabularies [36]. It allows collaborative work and enables standardized and consistent metadata within experiments. As a free usable Google Spreadsheet [37] plugin, OntoMaton supports researchers in a familiar environment and can be integrated into any layout. In addition to semantic annotation features, OntoMaton can support the ontology development process [21]. The Google spreadsheet environment allows, for both import and export formats, Microsoft Excel, OpenDocument, PDF, web page (HTML), comma separated values (CSV) and tab separated values (TSV).

*eleMAP*

eleMAP (for data element mapping) is a web services-based tool developed within the Electronic Medical Records and Genomics (eMERGE) Network [20]. The eMERGE Network was funded by the National Human Genome Research Institute (NHGRI). One of the goals is to develop tools to facilitate the harmonization of phenotypic data dictionaries for interoperable representations of phenotype data [20]. eleMAP allows semi-automatic mapping of data elements to standardized biomedical vocabularies (as NCI-T and SNOMED CT) and metadata registries (as caDSR). A RESTful interface that queries the caDSR (Cancer Data Standards Registry and Repository) can be used to reuse mappings. Additionally, the NCBO BioPortal REST service can be used for identifying concepts, mapping them to data items and submitting the mappings to the caDSR metadata repository. The mappings can be exported as Microsoft Excel or XML files [20].

*CEDAR Workbench*

The Center for Expanded Data Annotation and Retrieval (CEDAR) is an ecosystem for creating and refinement of biomedical metadata [38]. Central aspects are metadata templates to support researchers to upload annotated datasets to online repositories. The CEDAR Workbench is a tool set for creating and reusing metadata templates under consideration of the FAIR data principles. A terminology service component in the modular CEDAR architecture is used to semantically enrich metadata with terms from NCBO BioPortal ontologies. The interactive lookup service allows annotation of templates and fields during template design. Values of the fields can also be specified with ontology terms. CEDAR metadata is available in JSON, JSON-LD and RDF formats. The web service is open-access after registration. [39].

*Semantic Annotation Prototype of Wiktorin*

This service developed by Wiktorin is a collaborative tool for semantic annotation of medical data with the SNOMED CT, LOINC and ATC terminology standards [22]. The annotation process is supported by providing existing and similar mappings and cross-mappings of the UMLS metathesaurus. Interfaces for terminology integration are not provided. Since this tool is still in prototype development, the source code and information about import and export formats are not available.

*Metadata-Enricher*

Bernasconi et al. proposed an ontology-driven metadata enrichment workflow for genomic datasets [40] that was built within the GeCo (Data Driven Genomic Computing) project [41]. Metadata of genomic datasets is annotated with ontological terms, their preferred labels, synonyms, hypernyms and hyponyms [40]. Annotations are gathered from the EMBL-EBI Ontology Lookup Service [29]. Bernasconi et al. also presented an ontology selection method that precedes annotation. They designed a scoring system based on the measure of how well a term matches a value from the ontology. The automatically produced annotations are send to data curators if they are below a threshold. Data curators can manually select a suggested annotation, propose a new annotation or correct annotations [40]. Import and export formats are not documented.

*SAP*

The Semantic Annotation Pipeline (SAP) was developed to automate semantic annotation of metadata from public data repositories [42]. It is based on the CEDAR system and transforms the metadata from repositories into the CEDAR JSON-LD format before adding annotations to the CEDAR-formatted metadata. The metadata are annotated based on the CEDAR template. The named entity recognition tool Apache UIMA ConceptMapper is used for mapping metadata to ontology terms from NCBO BioPortal. SAP was published as a poster [42]; therefore, only limited information is available.

*D2Refine*

D2Refine (for Data Dictionary Refine) is a web-based platform for standardization and harmonization of clinical research study data dictionaries [24]. It was built on top of the open-source tool OpenRefine (formerly Google Refine) [43]. OpenRefine is a tool used to clean and transform large datasets in a spreadsheet-like interface. D2Refine leverages the extensive support of import formats of OpenRefine, such as CSV, MS Excel, XML and JSON; and database content, for instance, from the Database of Genotypes and Phenotypes (dbGaP)[44], the Phenotype Knowledge Base (PheKB) [45] and The Cancer Genome Atlas (TCGA) [46]. Besides various export formats (CSV, TSV, HTML, Excel, ODF, XML, RDF), D2Refine extends the export mechanism of OpenRefine to serialize models into standard representations such as OpenEHR's Archetype Definition Language (ADL), OMG Archetype Modeling Language (AML), W3C Shape Expressions (ShEx) and HL7 FHIR Profiles. D2Refine also extends the reconciliation service of OpenRefine to standardize data dictionary variables. Annotation suggestions based on the Common Terminology Services 2 (CTS2) implementation are offered and can be associated with the corresponding variable. Manual search can be used for alternate annotations. D2Refine can use all CTS2-compliant terminology services for suggestion and manual search.

### 4.2. Requirements Revisited

In this section we compare the found software system against the basic requirements which are summarized in Table 1. The comparison is described below in detail and summarized in Table 2.

#### 4.2.1. Requirement 1: Open Accessible and Open Code

Six (CEDAR Workbench, Rightfield, Swate, OntoMaton, Metadata-Enricher and D2Refine) out of ten reviewed semantic annotation services provide source code under an open license. The services ODMedit and CEDAR Workbench can be directly used after free registration. No online service nor source code was available for the prototype of Wiktorin, eleMAP and SAP.

**Table 2.** Comparison of semantic annotation services for requirements 1–7.

| Service | Requirement 1: Open Accessible/Open Code | Requirement 2: Common Standard Formats, Import/Export | Requirement 3: FAIR Terminologies | Requirement 4: Terminology Search | Requirement 5: Annotation Suggestion | Requirement 6: Interface for Terminology Service Integration | Requirement 7: Terminology Upload |
|---|---|---|---|---|---|---|---|
| ODMedit [32] | Yes */No | ODM/ODM | MDM repository (consists of UMLS codes) | Yes (MDM repository) | Yes | No indication | No indication |
| Rightfield [35] | Yes/Yes | MS Excel/MS Excel, CSV, RDF | NCBO BioPortal, Local Files | Yes | No | No indication | Yes, import of local files |
| OntoMaton [21] | Yes/Yes | MS Excel, OpenDocument, PDF, HTML, CSV, TSV | NCBO Bioportal, Linked Open Vocabularies, EBI Ontology Lookup Service | Yes | No | No indication | No indication |
| eleMAP [20] | No/No indication | Unknown/MS Excel, XML | caDSR, NCI-T, SNOMED CT | Yes | Yes | No indication | No indication |
| CEDAR Workbench [38] | Yes */Yes | XML/JSON, JSON-LD, RDF | NCBO BioPortal | Yes | Yes | No indication | No |
| Prototype Wiktorin | No **/No | No indication | SNOMED CT, LOINC, ATC | Yes | Yes | No indication | No |
| SAP [42] | No **/No | Unknown | NCBO BioPortal | Unknown | Yes | No indication | No indication |
| Metadata-Enricher [40] | No ′/Yes | Unknown | OLS | Unknown | Yes | No indication | No indication |
| Swate [23] | Yes/Yes | MS Excel/MS Excel, CSV, RDF | Chemical Entities Of Biological Interest, Environment Ontology, Gene Ontology, PSI-MOD, Proteomics Standards Initiative Mass Spectrometry vocabularies, Ontology for Biomedical Investigations, Phenotype and Trait Ontology, nfdi4pso, Plant Experimental Conditions, Plant Ontology, Relation Ontology, Plant Trait Ontology | Yes | No | No indication | No, but concepts can be added to the Swate ontology |
| D2Refine [24] | Yes /Yes | CSV, TSV, HTML, Excel, ODF, XML, RDF, ADL2.0 (OpenEHR RM or OpenCIMI RM) | Common Terminology Services 2 (CTS2) | Yes | Yes | (CTS2)-compliant terminology services | Yes, CTS2 compliant terminology |

* Registration, ** Not published, ′ Available as GitHub repository.

### 4.2.2. Requirement 2: Support for Common Data Formats

Rightfield, Swate and OntoMaton are plugins for Microsoft Excel [34] or Google Spreadsheet [37] and therefore only support these formats. ODMedit, as the name suggests, is intended for the standard ODM format. CEDAR Workbench provides a whole ecosystem for handling biomedical metadata, and supports caDSR [47] encoded XML files as input and JSON, JSON-LD and RDF as output formats. Similarly, eleMAP allows one to export Microsoft Excel or XML files. No information on supported input formats is given. D2Refine and its basis OpenRefine support a wide range of formats, including general formats such as CSV, and domain specific ones such as HL7 FHIR profiles. For the tools Prototype of Wiktorin, SAP and Metadata-Enricher, no information on supported formats is given.

### 4.2.3. Requirements 3 and 4: FAIR Terminologies and Interfaces for Terminology Service Integration

The services Rightfield, OntoMaton, CEDAR Workbench and SAP use the NCBO Bioportal to retrieve ontologies for semantic annotation. The means changes to the address of the connected BioPortal service are not described in detail in the corresponding publications. Due to time constraints, we did not analyze the available code base. All ontologies in the NCBO Bioportal are publicly available; whether they meet the requirements of the FAIR principles has to be checked individually. The metadata encoder integrates the EMBL-EBI Ontology Lookup Service (OLS) for biomedical ontologies [29]. Means to change the address of the connected OLS service are also not prominently described in the corresponding publication. Whether the ontologies contained in OLS meet the FAIR requirements must also be checked individually. The ODMedit, eleMAP and prototype Wiktorin software systems focus on medical ontologies/terminologies such as SNOMED CT, LOINC and NCIT. No connection to an external service is described for those systems. D2Refine comes with a default terminology implementation of NCI LexEVS CTS2 Services [48] for annotation suggestions. The service can be exchanged or complemented by any CTS2-compliant terminology service.

### 4.2.4. Requirement 5: Terminology Search

The terminology search function is a common feature of almost all services analyzed, with the exception of SAP and Metadata-Enricher (data not provided). While the other services provide the ability to change the set of ontology/terminology terms, ODMedit is limited to the terms in its public metadata repository, and further links for a UMLS search to the UMLS website are provided. More details on the requirements for extending to semantic concepts are described in the next section.

### 4.2.5. Requirement 6: Annotation Suggestion

Since searching and selecting appropriate semantic terms for a data element is labor intensive, automatically suggesting these terms is a valuable feature. Seven of the ten tools support this feature; Rightfield, OntoMan and Swate do not. The others integrate various techniques. SAP uses Apache's UIMA ConceptMapper to map metadata text to ontology concepts [42]. The Metadata-Enricher defines a "match score" based on the Needleman–Wunsch algorithm [49] to compute suggestions. Similarly, D2Refine calculates a similarity score based on the Levenshtein Algorithm of matching of the variable and its occurrence in the concept name or description for their default terminology [24]. eleMAP performs exact string matching followed by approximate search by normalizing the search string [20]. ODMedit suggests annotations using a method described by Christen et al. [50].

### 4.2.6. Requirement 7: Terminology Upload

Rightfield can be customized by loading local ontology files, and Swate allows adding custom concepts and ontologies. D2Refine can be complemented with Common Terminology Services 2 (CTS2)-compliant terminology services. For all other tools, no information about this opportunity is available.

## 5. Discussion

This review has provided an overview of semantic metadata annotation services in the biomedical field. For this purpose, we reviewed over 200 research articles retrieved via PubMed and Google Scholar. Most of the articles, although meeting our search criteria, had to be excluded from the review because they were not relevant. The criterion for inclusion of a publication in this review was the description and provision of a service for semantic metadata annotation. Thus, only articles describing a standalone service for linking metadata elements to semantic concepts were considered.

A number of services consist of components that provide supporting services, such as offering semantic concepts for enrichment/annotation or linking a text snippet within a data object to the corresponding semantic code (see Section 3). Such supporting services play an important role in the development of a semantic metadata annotation service, as formulated in Requirements 3 (FAIR terminologies), 4 (terminology search) and 5 (annotation suggestion). However, named entity recognition methods for text mining in particular are an extensive topic, and reviewing those components is beyond the scope of this review.

Furthermore, there are ontology-dependent web-services that integrate the display of semantic concepts and their associations with text fragments (e.g., [29,51]). They can be seen as a complement to ontology/terminology services rather than standalone services because they only accept unstructured text as input and do not involve the human in the loop.

Based on these criteria, ten relevant articles/software systems were identified and summarized in this work. In addition, we analyzed the extent to which these systems meet the seven basic requirements we have defined in the context of the research data infrastructure project NFDI4Health. Annotation services are needed to support researchers in using standard data formats and semantic annotations towards external terminologies. Annotation suggestions can support the annotation process in this regard. Furthermore, such services should be openly accessible and should provide access to FAIR terminologies for annotation. Finally, in order to meet the needs of different research areas and to integrate the services into already established methods and workflows, a range of formats and terminologies should be supported.

No tool met all the requirements completely. The annotation services Rightfield, CEDAR Workbench, OntoMaton, SAP and Metadata-Enricher provide interfaces to ontology portals (NCBO Bioprotal or OLS) and have in such a way a broad coverage. However, they do not seem to have interfaces for additional terminology integration. Only Rightfield and D2Refine provide the opportunity to upload local files, whereby D2Refine limits the upload to CTS2-compliant terminologies.

ODMedit, Swate, eleMAP and Prototype Wiktorin are tailored to specific use cases and also do not seem to have interfaces for additional terminology integration. However, the Swate terminologies could be adapted by Swate administrators. Rightfield and OntoMaton are designed for Microsoft Excel and Google Spreadsheets, respectively, and are not capable of handling other formats currently. Several tools offer annotation suggestions: either straightforwardly by providing similar annotations or by performing string matching (ODMedit, eleMAP and Prototype Wiktorin) or in a more complex way by using a concept recognition tool (SAP). SAP and eleMAP appear to be promising in terms of annotation suggestions, although neither has released code yet and neither is publicly available. ODMedit's approach to annotation suggestions allows for consistent semantic annotation based on UMLS concepts already used in other MDM-portal questionnaires. Whereas this is a great advantage for already existing concepts, it makes the annotation of new concepts cumbersome.

Another service under current development is Snap2Snomed [52]. This tool allows the collaborative creation and maintenance of simple maps to SNOMED CT. Due to lack of public documentation at the time of writing this review, it was not included.

A more complex requirement for semantic enrichment/annotation that was not explicitly considered in this review is the association of multiple semantic concepts to a digital

object. Several terminologies/ontologies and their backing technologies (OWL) allow one to construct more complex semantic classes by combining and intersecting existing concepts. Examples are the food classification standardization FoodEx2 [53] or the clinical terms terminology SNOMED CT [54]. A simple SNOMED example is the boolean AND concatenation of the two concepts "cyst" and "kidney" to describe a "renal cyst." However, as this review showed, even the most simple case (one-to-one correspondence) is not yet well supported by current software systems.

In addition, establishing frameworks for using machine readout for automatic metadata generation was not addressed in this review. An example of such a project is Health-SOS, where ECG data annotations were generated automatically in a real time setting [55,56]. Such developments, which are being pursued in many approaches and projects, show promise for making retrospective annotation obsolete in the future. However, even if all newly generated data are enriched with semantic codes during creation, the large body of existing data must also be transformed in order to be FAIR. This is especially true for longitudinal data and studies.

Based on this review, we conclude that there are already promising tools for semantic annotation systems, but no tool yet meets all requirements. Since interoperability of data inventories, questionnaires and data collection instruments is key to reusing and combining results from independent research studies, further research and software development in this area is needed.

**Author Contributions:** Conceptualization, J.S., J.D. and J.F.; methodology, J.S., J.D. and J.F.; validation, J.S., J.D. and J.F.; formal analysis, J.S., J.D. and J.F.; investigation, J.S.; resources, J.S., J.D. and J.F.; data curation, J.S.; writing—original draft preparation, J.S., J.D. and J.F.; writing—review and editing, J.S., J.D. and J.F.; visualization, J.S., J.D. and J.F.; supervision, J.F.; project administration, J.S., J.D. and J.F.; funding acquisition, J.F. All authors have read and agreed to the published version of the manuscript.

**Funding:** This work was done as part of the NFDI4Health Consortium (www.nfdi4health.de; accessed on 29 December 2021). We gratefully acknowledge the financial support of the Deutsche Forschungsgemeinschaft (DFG, German Research Foundation)— Project number 442326535.

**Institutional Review Board Statement:** Not applicable.

**Informed Consent Statement:** Not applicable.

**Data Availability Statement:** No new data were created or analyzed in this study. Data sharing is not applicable to this article.

**Acknowledgments:** This work was done as part of the NFDI4Health Consortium and is published on behalf of this Consortium (www.nfdi4health.de; accessed on 29 December 2021). We would like to thank Carina Vorisek (Berlin Institute of Health (BIH)), Sophie Klopfenstein (Berlin Institute of Health (BIH)), Matthias Löbe (Inst. for Medical Informatics (IMISE)) and Carsten Oliver Schmidt (UM Greifswald) for their contributions during the requirement specification and Carsten Oliver Schmidt for the contribution during review and editing.

**Conflicts of Interest:** The authors declare no conflict of interest.

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
