# Peer review of "Semantic Metadata Annotation Services in the Biomedical Domain—A Literature Review"

_applsci, doi:10.3390/app12020796_

Round 1

Reviewer 1 Report

This study aimed to review the Semantic Metadata Annotation Services in the biomedical domain. Although this review article lacks appropriate references and discussions, I have the following suggestions.

  • Authors must provide the background studies of biomedical Semantic Metadata Annotation and their application in the introduction section with enough references.
  • Which novelty do authors claim for this review? Authors should add the main scientific contribution of this article in a bulleted form at the end part of the Introduction section.
  • The major drawback of this review is the lack of reference and citation. Reference needs to be improved.
  • Authors need to describe the definitions based on references in a review article, not their own opinion. For example, an inappropriate statement in lines 67-68 “In our understanding, a metadata annotation…………..69 semantic codes”. A similar kind of statement without proper reference should be corrected.
  • Authors only list up a few tools/software in results based on PRISMA diagram. The authors should discuss the strength and weaknesses of other Semantic Metadata Annotation applications in the discussion section.
  • Authors should provide a conceptual figure of the application of Semantic Metadata Annotation service in the biomedical domain.
  • Authors should improve references mentioning the potential application of Semantic Metadata Annotation in several biomedical applications, For example, Ontology-based knowledgebase systems were studied in disease prediction and diagnostics, such as https://doi.org/10.1109/ACCESS.2021.3109806, https://doi.org/10.1109/ACCESS.2020.3040437, https://doi.org/10.1093/nar/gkz389.
  • Discussion sections need to be improved. Authors should discuss with case studies of biomedical Semantic Annotation, find opportunities/drawbacks current Semantic Metadata Annotation tools and make directions to improve the future annotation Services.
  • From the writing point of view, the manuscript needs to be checked for typos and the English language should be improved.

Reviewer 2 Report

The review titled ‘’Semantic Metadata Annotation Services in the biomedical domain - A Literature Review’’ presents an overview of semantic metadata annotation services in the biomedical field.  Nine relevant software systems were summarized and evaluated against the established requirements and the authors concluded no system meets the requirements despite active research on semantic annotation systems.

The aspect of the manuscript is quite extensive, although my main concern is about the organization of the article. As methodology and results are including, it is not like a typical Review.

What aspects of these metadata annotation services have actually been tested in other studies; The authors could also stress the limitations of the proposed approach.

Discussion Section is too short and should be extended, otherwise authors should be named as Conclusions.

Table 1 does not sound clear, please reorganize accordingly.

Round 2

Reviewer 1 Report

Thank you for addressing the comments.